# Obesity and Cancer: A Current Overview of Epidemiology, Pathogenesis, Outcomes, and Management

**DOI:** 10.3390/cancers15020485

**Published:** 2023-01-12

**Authors:** Sukanya Pati, Wadeed Irfan, Ahmad Jameel, Shahid Ahmed, Rabia K. Shahid

**Affiliations:** 1College of Medicine, University of Saskatchewan, Saskatoon, SK S7N 5E5, Canada; 2Royal College of Surgeon, Dublin 2, Ireland; 3Saskatoon Cancer Center, Saskatchewan Cancer Agency, Saskatoon, SK S7N 4H4, Canada

**Keywords:** obesity, cancer, malignancy, overweight, body mass index, epidemiology, cancer outcomes, cancer risk, treatment

## Abstract

**Simple Summary:**

Obesity has been linked to several common cancers including breast, colorectal, esophageal, kidney, gallbladder, uterine, pancreatic, and liver cancer. Obesity also increases the risk of dying from cancer and may influence the treatment choices. About 4–8% of all cancers are attributed to obesity. The underlying mechanism of obesity causing cancer is complex and is incompletely understood. Lifestyle changes that include diet, exercise, and behavior therapy are the mainstay of interventions. Drug therapy and weight reduction surgery result in a more rapid weight loss and may be used for a subgroup of cancer survivors with obesity. This review highlights the epidemiology and the risk associated with the development and recurrence of cancer in obese individuals and the management of obesity.

**Abstract:**

Background: Obesity or excess body fat is a major global health challenge that has not only been associated with diabetes mellitus and cardiovascular disease but is also a major risk factor for the development of and mortality related to a subgroup of cancer. This review focuses on epidemiology, the relationship between obesity and the risk associated with the development and recurrence of cancer and the management of obesity. Methods: A literature search using PubMed and Google Scholar was performed and the keywords ‘obesity’ and cancer’ were used. The search was limited to research papers published in English prior to September 2022 and focused on studies that investigated epidemiology, the pathogenesis of cancer, cancer incidence and the risk of recurrence, and the management of obesity. Results: About 4–8% of all cancers are attributed to obesity. Obesity is a risk factor for several major cancers, including post-menopausal breast, colorectal, endometrial, kidney, esophageal, pancreatic, liver, and gallbladder cancer. Excess body fat results in an approximately 17% increased risk of cancer-specific mortality. The relationship between obesity and the risk associated with the development of cancer and its recurrence is not fully understood and involves altered fatty acid metabolism, extracellular matrix remodeling, the secretion of adipokines and anabolic and sex hormones, immune dysregulation, and chronic inflammation. Obesity may also increase treatment-related adverse effects and influence treatment decisions regarding specific types of cancer therapy. Structured exercise in combination with dietary support and behavior therapy are effective interventions. Treatment with glucagon-like peptide-1 analogues and bariatric surgery result in more rapid weight loss and can be considered in selected cancer survivors. Conclusions: Obesity increases cancer risk and mortality. Weight-reducing strategies in obesity-associated cancers are important interventions as a key component of cancer care. Future studies are warranted to further elucidate the complex relationship between obesity and cancer with the identification of targets for effective interventions.

## 1. Introduction 

Obesity is a common disease and has been associated with several major chronic illnesses including coronary artery disease, diabetes mellitus, hypertension, osteoarthritis, sleep disorders, and mental illnesses, among others [1,2,3]. More importantly, there is growing evidence that obesity increases the risk of several solid-organ and hematological malignancies [4,5,6,7,8]. Obesity has been linked to cancer of the breast, colon and rectum, esophagus, stomach, gallbladder, uterus, pancreas, and ovary, among others [9,10,11,12,13,14,15,16,17,18,19,20,21,22,23,24,25]. The relationship between obesity and cancer is rather complex. Obesity is not only associated with an increased risk of cancer but may also increase the risk of cancer recurrence and mortality in cancer survivors [26,27]. Hence, the management of obesity as an early intervention in patients with early-stage cancer is important to improve cancer outcomes. 

Over the past decade, new evidence of the association between obesity and cancer risk and inferior outcomes has arisen [6,7,8,26,27]. New drugs that are more effective and relatively safer than the older agents have been approved for weight management in adults with obesity [28,29,30]. An increasing number of prospective studies are underway to address various aspects of obesity in cancer patients. This review was undertaken to provide readers updated information on obesity and cancer. The paper first describes the definition of obesity, the epidemiology of obesity and cancer in the general population, the use of new technology in measuring obesity and body fat distribution, and potential biologic mechanisms of the development of cancer in obese individuals. It then reviews recent data on the negative relationship between obesity and cancer outcomes, including the risk of recurrent disease and treatment-related complications. Then, it evaluates the management of obesity using various strategies in cancer survivors where weight reduction is an important adjunct to cancer treatment in reducing the risk of recurrent disease and a new obesity-related primary cancer. It ends with summarizing ongoing clinical trials related to obesity in cancer survivors. 

## 2. Methods

A literature search was performed using PubMed and Google Scholar. In addition, clinical practice guidelines, position papers by professional societies and organizations, and the reference lists of relevant articles were reviewed for identification pertinent articles. Due to the wide scope of this paper, a formal literature search was deemed impractical. The search was mostly limited to papers published in English over the past decade up to September 2022 and was focused on studies that investigated epidemiology, the pathogenesis of cancer in obesity, cancer incidence and the risk of recurrence in obesity, and the management of obesity. As this review addresses several aspects of cancer and obesity, multiple searches using various key words in combination with “obesity” and “cancer” were performed independently to identify key papers relevant to a section. The senior authors reviewed the citations and content of each section and made appropriate changes. Most clinical evidence is derived from meta-analyses and or systematic reviews of observational studies or pooled analyses of clinical trials.

## 3. Definition and Measurement of Obesity 

Obesity is characterized by the presence of excess body fat [1]. “Obese” and “obesity” are the terms used by clinicians to describe those individuals with a substantially increased risk of ill-health due to the accumulation of excessive body fat. Although obesity was recognized as a disease by the World Health Organization (WHO) in the late 1940s, it was only during the latter part of the century that the term became significant, after its connection was established with serious illnesses [31]. The WHO has defined obesity as “abnormal or excessive fat accumulation to the extent that health may be impaired” [1]. It results from a complex interplay between genetic, environmental, socioeconomic, and behavioral factors [32]. Body mass index (BMI) is the surrogate measurement of excess body fatness. Obesity is commonly described as a BMI or weight to height ratio of greater than or equal to 30 kg/m^2^ [1]. Individuals with a BMI of 25–29 kg/m^2^ are considered overweight. BMIs > 30 kg/m^2^ are further classified as follows: class I, a BMI of 30.0–34.9 kg/m^2^; class II, a BMI of 35.0–39.9 kg/m^2^; and class III, or extreme obesity, BMI ≥ 40 kg/m^2^. Obesity is also referred to as a high waist–hip ratio [33,34,35]. Abdominal obesity is defined as a waist–hip ratio above 0.90 in men and above 0.85 in woman [36]. Evidence supports the superiority of a high waist–hip ratio over BMI or waist circumference in predicting cardiometabolic risk [30]; waist circumference and waist–hip ratio have been found comparable to BMI and a better predictor of cancer risk [36,37,38]. 

Anthropometric measures are commonly used in epidemiological studies for body fat measurements; however, they have some limitations. In addition to anthropometric measurements, various other techniques have been used to measure body fat composition and distribution [39,40,41,42,43,44]. Dual energy X-ray absorptiometry (DEXA) is one of the standard tests for the direct measurement of whole-body fat and its regional distribution. A DEXA measurement of body fat > 25% in men and >30% in women is considered abnormal [39]. The other techniques include bioelectrical impendence, ultrasound, computed tomography (CT) scans, and magnetic resonance imaging (MRI) scans [40,41,42,43,44]. These methods and their potential advantages and disadvantages are described in Table 1.

## 4. Epidemiology 

Obesity has nearly tripled worldwide since 1975 [1,45]. In 2020, 39 million children under the age of five were either overweight or obese. If current trends continue, it is predicted that by 2025, 2.7 billion adults would be overweight, over 1 billion would be obese, and 177 million will be extremely obese, while approximately 38% of the world’s adult population will be overweight by 2030 and another 20% will be obese [45]. A meta-analysis of 230 studies involving more than 30 million individuals showed that excess body weight is associated with an increased risk of all-cause mortality [26]. Globally, it is estimated that obesity attributes to about 4–8% of all cancers with a range of <1% in low-income countries to 7–8% in high-income countries [5,7]. In the United States, it is estimated that obesity accounts for about 4.7% of new cases of cancer in men and 9.6% new cases of cancer in women [8]. However, the proportion of cancer linked to excess body weight varies according to the underlying cancer, with 51% of all liver or gallbladder cancers and 49.2% of all endometrial cancers in women and 48.8% of all liver or gallbladder cancers and 30.6% of all adenocarcinomas being attributed to obesity [8]. It is estimated that about 21% of all cancers that are linked to obesity could be prevented if the American adult population had a BMI < 25 or a healthy body weight [46]. Among those cancer survivors who are aged 20 years or older, 33.6% have been shown to be obese and 35.6% are reported to be overweight [47].

## 5. Biological Relationship between Obesity and Cancer

Obesity is a well-established predisposing factor for many malignancies, most notably breast and colorectal cancer [4,5,6,7,8]. Adipocyte tissue and its microenvironment may play a role in carcinogenesis, the development of metastases, and the progression of the disease [48]. However, the underlying mechanism resulting in carcinogenesis is complex and not yet fully understood. Altered fatty acid secretion and metabolism, extracellular matrix remodeling, the secretion of anabolic and sex hormones, immune dysregulation, chronic inflammation, and changes in the gut microbiome have been linked to carcinogenesis, the development of metastases, and the progression of cancer in obesity [49,50,51] (Figure 1). It is likely that different mechanisms lead to the development of different cancers.

There are three primary biological mechanisms that have been proposed as a risk of the development of malignancy in obese individuals [52]. The first of these mechanisms is related to the concept of adipose tissue acting as an “organ”, with the capacity to release chemical mediators and enzymes. Specifically, this mechanism describes increased synthesis of estradiol from androgens due to the presence of aromatase found in peripheral adipose tissue [52,53,54]. The overproduction of estrogen by adipose tissue has been linked to an increased risk of developing breast, endometrial, ovarian, and other cancers [55].

The second mechanism is the consequence of hyperinsulinemia due to increased BMI, which stimulates the normal growth function of insulin as well as prolonging the duration of the action of insulin-like growth factor-1 (IGF-1). Insulin and IGF-1 blood levels are frequently higher in obese people. Insulin resistance, another established cancer risk factor, causes high amounts of insulin, or hyperinsulinemia, which occurs before the onset of type 2 diabetes [49,52,56]. The development of colon, renal, prostate, and endometrial cancer may be aided by high insulin and IGF-1 levels [56].

The third part of the mechanism relates to the proinflammatory environment cultivated by the altered secretion of many adipokines (polypeptide hormones) by adipose tissue, specifically increased levels of leptin, which is a potent inflammatory, proliferative, and anti-apoptotic agent [52,57]. Adiponectin, which is another adipokine that has antiproliferative properties, is low in obese individuals with a healthy weight [49]. Excess fat tissue leads to adipocyte hypertrophy and cell death resulting in the chronic, subclinical inflammation of adipose tissue [57]. Several pre-clinical studies support that chronic inflammation in adipose tissue triggers carcinogen and the progression of cancer [58]. Individuals with excess body weight have altered levels of inflammatory cytokines including IL-6, TNFα, and C-reactive protein [59]. The prevalence of chronic inflammatory diseases, including gallstones and non-alcoholic fatty liver disease, is higher in obese adults. These factors create oxidative stress, which damages DNA and makes people more likely to develop biliary tract, liver, and other malignancies [60]. Obesity may also increase the risk of cancer by reducing tumor immunity and altering the mechanical characteristics of the tissue that surrounds growing tumors [61]. Adipokines, immune cell modulation and systemic inflammation, angiogenesis, metabolic changes, extracellular matrix modulation, and extracellular vesicles such as exosomes have been implicated in metastases [50].

### Obesity, Body Fat Distribution and Metabolic Syndrome

A higher BMI is strongly associated with metabolic syndrome, which includes higher fasting levels of insulin, glucose, triglycerides, and residual cholesterol, along with low levels of high-density lipoprotein (HDL) cholesterol [62,63]. The amount and location of adipose tissue are important factors that influence insulin resistance and dyslipidemia. Higher fatness has a negative metabolic impact that is already noticeable in late childhood and could worsen over time [64]. Additionally, increased body fatness increases systolic and diastolic blood pressure and lowers immunity due to its link to increased pro-inflammatory substances such as interleukin-6 [65].

Adults have a wide variety of body fat distribution in their body and BMI alone does not capture the complex association between excess fat and the risk of cancer and progression. Variation in the distribution of body fat is now understood to play a significant role in predicting the negative health effects of obesity with visceral fat carrying a higher risk versus subcutaneous fat [66]. Lower body (gluteofemoral obesity), upper body (abdominal obesity), and visceral fat stores all have distinct fatty acid metabolic properties [67,68]. The metabolic side effects of obesity are influenced by the selective dysregulation of these depots. For example, the adverse effects of fat distribution on health are significantly influenced by the dysregulation of upper body, non-splanchnic adipose tissue lipolysis. Type II diabetes mellitus, hypertension, insulin resistance, dyslipidemia, and early cardiovascular mortality have all been evidently linked to upper body obesity, particularly when visceral fat levels are elevated [67,69]. Furthermore, there is evidence that central obesity may be a better predictor of all-cancer risk than body size [37]. According to Mendelian randomization research, the highest metabolic risk factors for malignancies linked to fat are those with greater fasting insulin levels [70].

## 6. Obesity and Cancer Risk

Evidence suggests that weight gain during adulthood is associated with an increased risk of the development of post-menopausal breast, colorectal, endometrial, renal, and high-risk prostate cancers [4,5,6,7,8]. A large population-based study involving 5.24 million individuals with 166,955 new instances of 22 types of cancer supported strong links between BMI and the most common site-specific cancers [71]. According to studies utilizing Mendelian randomization, increased body fatness is associated with a greater chance of developing ovarian, esophageal, gastric, pancreatic, renal, colorectal, endometrial, and other cancers [72,73,74,75]. Mendelian randomization uses germline genetic variations that are strongly related to potentially modifiable exposures as proxies (or “instrumental variables”) for the target risk factor in order to assess causation in observational epidemiology. 

The International Agency for Research on Cancer (IARC) 2020 report found strong evidence for obesity and the risk of 13 different cancers [76]. Post-menopausal breast, colorectal, endometrial, esophageal, pancreatic, renal, liver, stomach, gallbladder, ovarian, thyroid, multiple myeloma, and meningioma are among the cancers that the obese population is more likely to develop (Table 2) [4,5,6,7,8,9,10,11,12,13,14,15,16,17,18,19,20,21,22,23,76,77,78]. The evidence of an association between obesity and cancers of the mouth, pharynx and larynx, prostate, and male breast, as well as diffuse large B-cell lymphoma, is moderate [76]. A high risk is shown for three of the most challenging cancers to treat—pancreatic, esophageal, and gallbladder—as well as the two most prevalent malignancies—breast and colorectal cancer. An umbrella review of 204 systematic reviews and meta-analyses investigated the relationship between excess body fat and cancer risk and reported a range from a 9% increase (RR 1.09, 95% confidence interval 1.06 to 1.13) for rectal cancer in men to a 56% (1.56, 1.34 to 1.81) increased risk of biliary tract system cancer with every 5 kg/m^2^ increase in BMI [79]. The risk of post-menopausal breast cancer in women who never received hormone replacement therapy was 11% (1.11, 1.09 to 1.13) with every 5 kg of weight gained in adulthood whereas the risk of endometrial cancer increased by 21% for each 0.1 increase in the waist–hip ratio (1.21, 1.13 to 1.29) [79].

The association of obesity with the risk of breast cancer is complex, with an inverse or neutral relationship between obesity and breast cancer diagnosed in pre-menopausal and a positive association in post-menopausal women, especially of hormone receptor-positive breast cancer [20,21,80,81]. There is evidence that an increased level of whole body fat measured by dual-energy X-ray absorptiometry in women with a normal BMI is associated with an elevated risk of breast cancer with an adjusted hazard ratio of 1.89 (95% CI, 1.21–2.95) for all invasive breast cancer and 2.21 (95% CI, 1.23–3.67) for hormone-receptor positive breast cancer; there is also a 56% increased risk of developing hormone receptor-positive breast cancer per 5 kg increase in trunk fat [81]. Insulin resistance, breast adipose inflammation, the increased expression of aromatase enzyme, elevated leptin, and the inflammation of breast adipose tissue have all been attributed to obesity-associated post-menopausal breast cancer [57,82]. There seems to be a gender-specific association between colorectal cancer and obesity in individuals with Lynch Syndrome. A meta-analysis of four studies involving individuals with Lynch Syndrome reported a two-fold risk of colon and rectal cancer in obese men compared with men with a healthy weight (Summary relative risk = 2.09; 95%CI: 1.23–3.55). No significant risk of colorectal cancer was noted in women. Individuals with an MLH1 germline mutation had a 49% risk of colorectal cancer compared to subjects with a healthy weight ((SRR = 1.49; 95%CI: 1.11–1.99) [83]. 

## 7. Obesity and Cancer Outcomes

Evidence supports the suggestion that obesity is not only associated with an increased risk of cancer but may also increase the risk of recurrence in early-stage cancer and be correlated with inferior outcomes [26,27,84,85,86]. Crosstalk between cancer cells and adipose tissue via hyperinsulinemia, proinflammatory cytokines, adipokines, and extracellular matrix proteins promotes metastases [50]. In particular, central obesity that results from the deposition of visceral adipose tissue has been linked to progression [50].

A systematic review and meta-analysis of 230 cohort studies showed that overweight and obesity have been associated with an increased risk of all-cause mortality [26]. Obesity has also been associated with inferior outcomes in some malignancies. Nevertheless, the relationship between obesity and cancer outcomes are not as clearly understood. The variability between each malignancy and even among subtypes lends to variability in the analysis of a causal relationship. A systematic review and a meta-analysis of 203 studies involving 6.3 million cancer patients showed that obesity was associated with an increased overall and cancer-specific mortality [27]. In general, obesity is associated with a 14% increased risk of overall mortality (hazard ratio [HR] for overall survival 1.14; 95% CI, 1.09–1.19; *p* < 0.001) and 17% increased risk of cancer-specific mortality (HR for cancer-specific survival, 1.17; 95% CI, 1.12–1.23; *p* < 0.001). Furthermore, obesity was associated with a 13% increased risk of recurrence (HR for recurrence, 1.13; 95% CI, 1.07–1.19; *p* < 0.001) [27] (Table 3). Obese patients with breast, colorectal, or uterine cancers had an overall poor survival. Obesity in patients with breast, colorectal, prostate, and pancreatic cancers was associated with high cancer-specific mortality and obese patients with breast, colorectal, prostate, and gastroesophageal cancers had high recurrence rates. Conversely, obese patients with melanoma, lung, and kidney cancer had better survival compared with non-obese patients [27].

The inferior outcomes in obese patients noted in certain cancers could be due to several factors, such as an underlying metabolic syndrome, hormonal factors related to some endocrine-dependent cancers, low physical activity, and undertreatment. Significantly, the practice of dose-capping in obese patients discussed below is relevant to all cancer types [87]. Moreover, other standard therapies have been associated with poorer outcomes in obese cancer patients. For example, surgical resections among obese patients have been shown to result in an increased incidence of post-operative complications including wound infection, prolonged operative time, and a greater risk of blood loss [88]. Evidence supports the suggestion that patients with gastrointestinal cancer following radical surgery have inferior long-term survival if they experience operative complications [89]. In addition, radiation therapy has been shown to have poorer results among obese patients; this has been attributed to the labor of daily setup along with the freedom of movement of the tumor within the fat of the patient [90].

### 7.1. Breast Cancer

Obesity not only increases the risk of recurrent breast cancer but also increases the risk of major comorbid illness and has a negative effect on breast cancer survivors’ quality of life [91,92,93]. A meta-analysis of 82 studies of breast cancer survivors showed that obesity increases the risk of breast cancer-specific and all-cause mortality by 35% and 41%, respectively [94]. Another meta-analysis of 13 studies involving 8944 women with triple negative breast cancer showed that women who were overweight had shorter disease-free survival (HR = 1.26; 95%CI: 1.09–1.46) and overall survival (HR = 1.29; 95%CI: 1.11–1.51) compared with women with normal-weight [95].

Evidence suggests that about 30–50% women gain more than 5% of their body weight during and following chemotherapy, which may persist for 5 years beyond diagnosis [96]. The onset of obesity in relation to diagnosis along with menopausal status confers different trends. Patients classified as obese (BMI > 30) 1 year prior to diagnosis are associated with increased breast cancer-specific mortality, in both pre- and post-menopausal cohorts. Patients who are obese at the time of diagnosis also presented with an increased mortality risk; however, it was less so than the pre-diagnosis studies [96]. Post-diagnostic obesity was shown to be a poor diagnostic marker for recurrence [87,97]. For example, an analysis of seven studies demonstrated that a weight gain of 5 kg within 6 months of diagnosis was a worse prognostic marker by 31% (95% CI: 0.97–1.75); the opposite trend was not significantly noted with weight loss [86]. Likewise, another systemic review and meta-analysis of 12 studies involving 23,932 breast cancer survivors showed that weight gain after the diagnosis of breast cancer is associated with a 12% higher risk of all-cause mortality compared with maintaining body weight [98]. The higher risk of mortality (HR of 1.23) was evident for a weight gain of ≥10.0%. There is also evidence of a negative effect of obesity on the efficacy of an adjuvant aromatase inhibitor in women with hormone receptor-positive breast cancer due to an increased peripheral aromatase inhibitor [97,99].

### 7.2. Colorectal Cancer

The prognosis of colorectal cancer has also been shown to be affected by the patient’s BMI. Studies have shown that obese patients have an increased predisposition to present with late-stage cancer (II or III) and with a greater number of lymph nodal metastases (N > 3) [100]. There is evidence that pre-diagnostic obesity is associated with an increased risk of disease-specific mortality and decreased overall survival [27,100,101]. Obesity has been linked to a 14% increase in both colorectal cancer-specific and all-cause mortality [102]. A meta-analysis of 16 prospective cohort studies that included 58,917 patients showed that being obese before cancer diagnosis was associated with a 22% increased risk of colorectal cancer-specific mortality and a 25% risk of all-cause mortality, while a BMI of ≥35 after diagnosis of colorectal cancer was associated with a 13% increased risk of all-cause mortality [103].

### 7.3. Pancreatic Cancer

In pancreatic cancer, both the loss of muscle mass and function or an increase in fat mass and obesity have been associated with poor outcomes [27,104,105]. Petrelli and others found that obesity was associated with a 28% risk of pancreatic cancer-related mortality [27]. Another systematic review and meta-analysis of 13 studies found that each one kg/m^2^ increase in BMI resulted in a 10% increase in mortality [105].

### 7.4. Endometrial Cancer

Obesity is not only a major risk factor for endometrial cancer but also correlates with inferior outcomes in women with endometrial cancer [27,106,107]. A high BMI and waist circumference before and after the diagnosis of cancer among endometrial cancer survivors are associated with inferior disease-free and overall survival [107]. A meta-analysis of 46 studies showed that obesity is associated with a 34% risk of all-cause mortality and a 28% risk of recurrence in women with endometrial cancer [106].

### 7.5. Prostate Cancer

Each 5 kg/m^2^ increase in BMI has been linked to a 21% increase in the probability of the biochemical recurrence of prostate cancer [108]. A systematic review and meta-analysis of 59 studies involving 280,199 patients showed that obesity increases the risk of prostate cancer-specific mortality by 19% and all-cause mortality by 9%. A 5 kg/m^2^ increase in BMI resulted in a 9% increase in prostate cancer-specific mortality and a 3% increase in all-cause mortality [109]. Delayed diagnosis, the presence of a biologically aggressive cancer and the lower rate of effective treatments including radical prostatectomy and positive resection margins have been attributed to inferior outcomes in obese men with prostate cancer [110].

## 8. Obesity and Treatment-Related Adverse Effects

Obesity increases several cancer treatment-related adverse effects [86,96]. Lymphedema is a complication of axillary lymph nodal surgery and radiation in women with breast cancer. The risk of lymphedema in women with breast cancer is several-fold higher compared with women with normal body weights (the odd ratio varies from 2.93 to 3.60) [111,112]. Likewise, chemotherapy-induced peripheral neuropathy is a common side effect of several anti-cancer agents and has been associated with impaired quality of life. Some cohort studies have shown higher rates of taxanes and platinum-related neuropathy in obese patients [113,114]. Recent evidence also highlights that excess body fat has been associated with a high risk of treatment-related cardiotoxicity [115,116]. A meta-analysis of 15 studies involving 8745 women with early-stage breast cancers who received adjuvant anthracyclines and trastuzumab showed that obesity was associated with a 47% increased risk of cardiac toxicities [115]. Obesity also increases radiation therapy-related toxicities [90,117]. A systematic review and meta-analysis of 38 studies involving 15,623 breast cancer survivors showed that a BMI of >25 was associated with an 11% risk of radiation-related acute dermatitis in breast cancer [117]. Several reports also support an association between high BMIs and surgical complications in cancer patients [118,119].

### Treatment Selection

Obesity may influence the treatment decision regarding specific types of cancer therapy [96]. Evidence shows that a high BMI in women with breast cancer does not negatively influence treatment decisions in relation to adjuvant chemotherapy but may affect decisions regarding immediate breast reconstruction following mastectomy [120,121]. More importantly, due to toxicity concerns, obese patients may receive a maximum (capped) dose that is lower than the full weight-based dose [122,123,124]. The practice of dose capping may result in a worse prognosis and inferior outcomes in obese cancer patients, especially in the context of adjuvant chemotherapy for early-stage cancer or definite treatment for highly chemosensitive cancers such as aggressive lymphoma [125,126]. The clinical practice guidelines advise against dose capping in obese patients and recommend for a full weight-based chemotherapy dose [86,127].

## 9. Management of Obesity in Cancer Survivors

Many cancer survivors experience weight gain following the diagnosis of cancer and its treatment [128,129]. As described above, obesity not only increases the risk of recurrence in some cancers but also increases the risk of diabetes mellitus, cardiovascular disease, and poor quality of life. Interventions directed at weight reduction are an important component of the survivorship care of overweight cancer patients [96,130]. The negative relationships documented between physical activity and the risk of several cancers, including some of the most prevalent types (breast, lung, bowel, and kidney), highlight the significance of an active lifestyle in cancer prevention, either through direct mechanisms, such as enhanced metabolic management, or through its involvement in preventing adult weight gain [131,132,133]. 

Evidence demonstrates that structured exercise combined with dietary support for weight loss results in greater weight loss than exercise or diet alone and has the greatest impact on blood-borne biomarkers linked to common cancers, such as insulin resistance and circulating levels of sex hormones, leptin, and inflammatory markers [134,135,136,137,138]. Many cancer guidelines recommend that survivors maintain a healthy weight, but there is a lack of evidence regarding which weight loss method to recommend [124,139,140]. Lifestyle interventions that include diet, exercise, and behavior therapy are the mainstay of interventions related to lifestyle modification (Figure 2). Pharmacological and surgical interventions are not well studied in cancer survivors. A Cochrane meta-analysis of 20 studies involving 2028 breast cancer survivors evaluated the effect of different body weight loss approaches in overweight or obese breast cancer survivors and found that weight loss interventions, especially multimodal interventions that include diet, exercise, and psychosocial interventions resulted in a reduction in body weight, body mass index, waist circumference, and better overall quality of life [141].

### 9.1. Diet

A consistent caloric deficit regardless of the type of diet results in weight loss [142,143]. The 2-year POUNDS LOST (Preventing Overweight Using Novel Dietary Strategies) trial randomized study participants into four diets groups with a deficit of 750 kcal per day and showed that there was no difference in the short- and long-term weight loss among the four groups [142]. However, macronutrient components of a low caloric diet can contribute to cardiometabolic outcomes [144]. The Diet and Androgen-5 (DIANA-5) trial examined whether a dietary adjustment based on macrobiotic and Mediterranean diet principles can lower the incidence of events associated with breast cancer. The early results showed that the DIANA-5 dietary intervention is effective in reducing body weight and metabolic syndrome parameters [145]. According to the Obesity Guidelines, an energy deficit of 500–750 kcal per day can result an average weight loss of 0.5–0.75 kg per week, which is 1200–1500 kcal per day for women and 1500–1800 kcal per day for men [146]. 

### 9.2. Exercise

Aerobic exercise on a regular basis improves physical fitness and endurance and in conjunction with diet not only has a positive effect on weight reduction but also reduces the risk of metabolic syndrome and cardiovascular complications by reducing visceral fat, blood pressure, and lipid levels and improving glycemic control [147,148]. As most of the published literature on weight reduction programs lacks long-term follow-ups, it is not known whether weight changes are sustained beyond the intervention periods.

The SUCCESS C (Docetaxel Based Anthracycline Free Adjuvant Treatment Evaluation, as Well as Lifestyle Intervention) phase 3 randomized trial assessed whether physical activity and healthy diets following adjuvant chemotherapy in 3643 women with early-stage breast cancer reduce disease-free survival [149]. Women in the intervention arm had a significant reduction in baseline weight compared with the non-intervention arm. Overall, 1477 women completed the 2-year lifestyle intervention program. An interim analysis showed that women who completed the 2-year lifestyle intervention program had a significantly better disease-free survival than those who did not complete (hazard ratio 0.35) [150]. Several trials are examining whether exercise with or without dietary intervention in obese or healthy weight cancer survivors could improve oncology outcomes (Table 4).

### 9.3. Behavior Therapy

Behavior therapy involves the use of various techniques and strategies to replace existing eating and activity behavior with a healthy eating and active lifestyle. It includes tracking eating, calorie intake, and physical activity, modifications of the environment to avoid overeating, the creation of exercise plans, and setting realistic goals, among others [96,130].

### 9.4. Drug Therapy

Currently, few drugs are approved for weight loss [28]. The most important of these are glucagon-like peptide-1 (GLP-1) analogue liraglutide and semaglutide. In a randomized phase 3 trial, when liraglutide was injected subcutaneously at a dose of 3.0 mg once-daily to 3731 patients as an adjunct to diet and exercise, it was associated with significant weight reduction. For example, after 56 weeks of treatment, 63.2% of treated patients vs. 27.1% with placebo lost at least 5% of their body weight and 33.1% of treated patients vs. 10.6% of the control group lost more than 10% of their body weight [29]. Semaglutide is available as a weekly injection [30]. A handful of preclinical studies suggest their association with thyroid and pancreatic cancer [151,152]. Other suggest that GLP-1 may have a protective effect on a reduction in the growth of certain cancers such as prostate and breast cancer [153,154,155]. Although there is paucity of research of the long-term safety of GLP-1 analogues in cancer survivors, it is a reasonable option in diabetic cancer survivors with excess body weight or those who are normoglycemic but failed other weight reduction measures. Other drugs that have shown efficacy in weight reduction include orlistat, the phentermine–topiramine combination, the bupropion–naltrexone combination, benzphetamine, phendimetrazine, and diethylpropion [28] (Table 4). However, unlike GLP-q analogues, they have more side effects, drug interactions, and several contraindications [28,156].

**Table 4 cancers-15-00485-t004:** Drugs with US Food Drug Administration (FDA) approved indication for obesity.

Drug [28,156]	Mechanism of Action	Dose	Weight Change Relative to Placebo	Side Effects
Semaglutide	glucagon-like peptide 1 receptor (GLP1R) agonists, decreases appetite and delays gastric emptying and gut motility	2.4 mg once per week subcutaneous injection	2.4% to 14.9%	abdominal pain, constipation, diarrhea, nausea, vomiting, and pancreatitis (rare)
Liraglutide	GLP1R agonists, decreases appetite and delays gastric emptying and gut motility	3.0 mg per day subcutaneous injection	2.6% to 8%	diarrhea, nausea, vomiting, constipation, dyspepsia, abdominal pain, pancreatitis, and gallstones
Naltrexone SR/bupropion SR	Opioid receptor antagonist/dopamine and noradrenaline reuptake inhibitor causing appetite suppression	32 mg/360 mg oral twice daily	dose dependent 1.3% to 6.1%	headaches, hypertension, sleep disorders, nausea, constipation, vomiting, diarrhea, palpitation, dizziness, tremor, and others
Orlistat	Pancreatic lipase inhibitor causing excretion of 30% of ingested trigylcerides in stool	120 mg 3 times daily	6.1% to 10.2%	nausea, diarrhea, steatorrhea, abdominal bloating, and hepatitis
Phentermine/ topiramate	Sympathomimetics/anticonvulsant causing appetite suppression	15 mg/92 mg once daily oral	dose depending 1.2% to 7.8%	tachycardia, xerostomia, constipation, headache, sleep disorder, anxiety, depression, suicidal ideation, and cardiovascular event
Phendimetrazine	Sympathomimetics causing appetite suppression	Short-term (≤12 weeks) 17.5 to 35 mg 2 or 3 times daily	Not available	hypertension, ischemic events, palpitations, tachycardia, headache, insomnia, overstimulation, psychosis, and others

### 9.5. Weight Reduction Surgery

Weight reduction surgery or bariatric surgery, such as sleeve gastrectomy or adjustable gastric banding, is typically considered in patients with a BMI > 40 kg/m^2^ or with a BMI of 35–40 kg/m^2^ with associated comorbid conditions such as obstructive sleep apnea [157]. Ongoing research has deemed that bariatric and related surgeries continue to be safe and will likely be used in the future [158]. The most common and considered to be highly effective forms of surgery used are the sleeve gastrectomy (SG) and Roux-en-Y gastric bypass (RYGB) [159]. Scant evidence suggests that the benefit of bariatric surgery as a weight reduction intervention is similar in cancer survivors compared with those with no history of cancer [160,161]. A systemic review and meta-analysis of six observational studies involving 51,740 patients demonstrated that bariatric surgery was associated with a 55% reduction in cancer risk. Patients who underwent bariatric surgery for obesity have a 27–59% lower risk of developing cancer than weight- and age-matched controls [162]. The advantages of bariatric surgery might only apply to malignancies linked to obesity, such as those of the breast and endometrium, where the average risk reduction is 38% (*p* = 0.0001). In contrast, the benefits of bariatric surgery are significantly more moderate (9%) among malignancies unrelated to obesity, such as those of the lung and bladder; this level of risk reduction is comparable to that of individuals who do not undergo bariatric surgery (*p* = 0.37) [163].

## 10. Future Directions

The pathogenesis of cancer development and recurrence in obesity is multifaceted and heterogeneous for different cancers and is also not fully understood. Further studies are warranted to fully understand the mechanisms specific to each cancer and to identify future targets for primary and secondary cancer prevention. While several studies have demonstrated a correlation between obesity and negative cancer outcomes, further studies are warranted using novel clinical and molecular markers. Some of the limitations of the current literature include the use of a fixed BMI threshold of 30 kg/m^2^ to distinguish between obese and non-obese individuals, insufficient information about the timing of obesity, and a lack of adjustment for psychosocial, genetic, environmental, and behavioral factors. Due to the limitations associated with anthropometric measurements, the routine use of novel methods that are more accurate in measuring body fat and its distribution are important for future studies in cancer and obesity.

Trials are ongoing globally to gain clarity on the relationship between weight reduction, exercise, and the reduction of the risk of cancer or its recurrence (Table 5). The Breast Cancer Weight Loss Study, or BWEL, is a randomized phase III trial that is examining the effectiveness of weight loss on invasive-disease-free survival in early-stage breast cancer survivors with a baseline BMI of less than 27 kg/m^2^ [164]. The LIVES (Lifestyle Intervention for Ovarian Cancer Enhanced Survival) trial will assess how diet and exercise affect the prognosis of women with advanced stage ovarian cancer [165]. LIVES will be the largest behavior-based lifestyle intervention trial for ovarian cancer survivors when it is finished. The CHALLENGE trial is currently investigating how moderate-intensity physical activity may reduce the risk of recurrence and mortality among colon cancer survivors [166]. Intense aerobic exercise and muscle building will be studied in the INTERVAL (Intense Exercise for Survival Among Men with Metastatic Castrate-Resistant Prostate Cancer) trial to see whether they can improve overall survival in men with advanced prostate cancer [167].

## 11. Conclusions

Obesity is a one of the major but preventable global health crises that has been linked to major chronic illness and several cancers, with increased morbidity and mortality. Among the various cancers, breast, colorectal, endometrial, esophageal, pancreatic, renal, hepatic, stomach, gallbladder, ovarian, and thyroid cancer, as well as multiple myeloma and meningioma, are strongly associated with obesity. The underlying mechanism of obesity causing cancer is incompletely understood and involves adipokines, inflammation, an altered extracellular matrix, altered fatty acid metabolism, and the secretion of insulin-like growth factors and estrogen. Weight-reducing strategies in obesity-associated cancers are important interventions as a component of cancer care in reducing cancer-specific and overall mortality in cancer survivors with excess body weight. Regular exercise and diet, in conjunction with behavior therapy, are the primary elements of weight reducing strategies. More data are needed regarding the efficacy and safety of pharmacologic and surgical interventions as a major weight reduction strategy in cancer survivors.

## Figures and Tables

**Figure 1 cancers-15-00485-f001:**
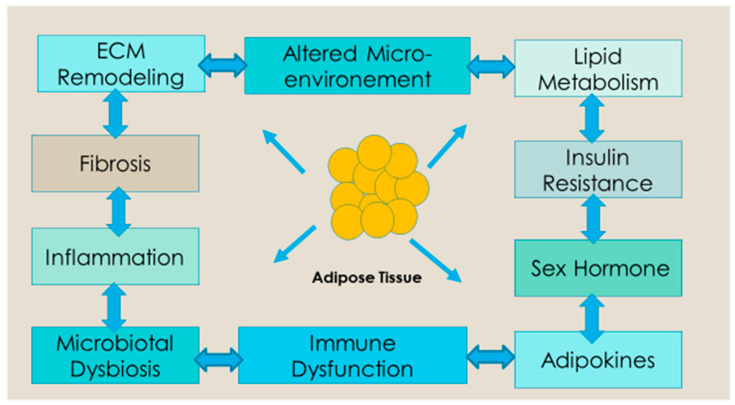
Potential mechanism of obesity-induced dysfunction of adipose tissue on tumor initiation, progression, and recurrence. Excess fat accumulation results in dysfunction of adipose tissue that causes increased production of proinflammatory cytokines, sex hormones, and lipid metabolites, along with impaired adipocyte-derived cytokines or adipokines profiles and insulin resistance. The altered adipose tissue is a source of ECM remodeling, fibrosis, cancer-associated adipocytes, impaired microbial metabolism, adipocyte progenitors, inflammation, and altered micro-environment. These factors contribute to tumor initiation, growth, and recurrence.

**Figure 2 cancers-15-00485-f002:**
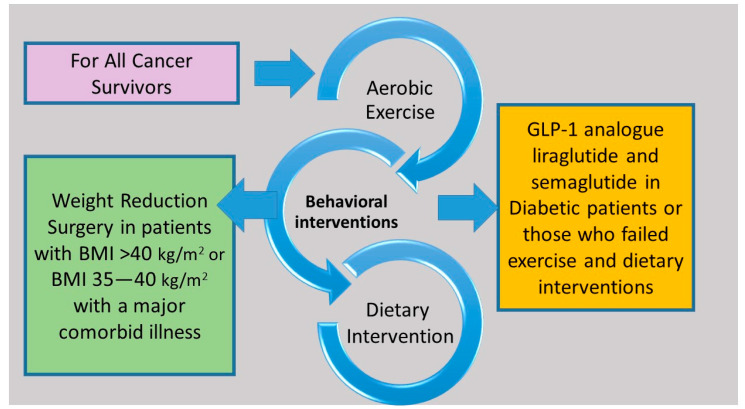
Structured exercise in combination with dietary support and behavior therapy are effective interventions for all cancer survivors. Treatment with glucagon-like peptide-1 analogues and bariatric surgery can be considered in selected cancer survivors.

**Table 1 cancers-15-00485-t001:** Methods for the estimation of body fat composition and its distribution.

Methods	Technical Principles	Potential Benefit and Drawbacks
Anthropometric measurement	Manual measurement of weight to height ratio (Body mass index), waist–hip ratio, waist circumference, arm circumference, skinfold thickness, etc.	Low cost and easy to measure but are not fully accurate or validated in different ethnicities
Dual energy X-ray absorptiometry (DEXA)	Generates X-rays at two different energies and uses differential attenuation of the X-ray beam at two energies to measure body composition including fat, fat free mass, and bone mineral density	Highly accurate, relatively low cost, and able to assess whole body but associated with low radiation exposure
Ultrasound	Reflection of ultrasound waves from tissue in the path of the beam	Highly accurate, no radiation exposure, but user-dependent and lack of standardization
Computed tomography (CT) scan	Uses X-ray beam and special digital X-ray detectors for generation of cross-sectional imaging with volumetric reconstruction of body adipose tissue and other tissues	Highly accurate and able to assess whole body but with greater radiation exposure and higher cost
Magnetic resonance imaging (MRI)	Employ powerful magnets, which produce a strong magnetic field that forces protons in the body to align with that field and generate cross-sectional imaging with volumetric reconstruction on the basis of different magnetic properties of water and fat	Provides better soft tissue contrast than other imaging studies and differentiates between fat, water, and muscle better; is able to assess the whole body but has a higher cost, limited access and longer scan times compared with CT and ultrasound
Bioelectrical impedance	Based on the principle that electric current flows at different rates through the body depending upon its composition	Low cost but limited accuracy

**Table 2 cancers-15-00485-t002:** Gender-specific summary of cancer risk for each 5 kg per m^2^ increase in BMI for major cancers with strong evidence of relationship with obesity.

Type of Cancer	Number of Cohorts	Relative Risk (95% Confidence Interval)	
		Women	Men
Endometrial cancer [4]	19	1.59 (1.50–1.68)	NA
Gallbladder cancer [4]	4	1.59 (1.02–2.47)	1.09 (0.99–1.21)
Esophageal adenocarcinoma [4]	5	1.51 (1.31–1.74)	1.52 (1.33–1.74)
Kidney cancer [4]	12	1.34 (1.25–1.43)	1.24 (1.15–1.34)
Postmenopausal breast cancer [4]	34	1.12 (1.08–1.16)	NA
Hpatocellular cancer [19]	9	1.12 (1.03–1.22)	1.19 (1.09–1.29)
Pancreatic adenocarcinoma [23]	23	1.10 (1.04–1.16)	1.13 (1.04–1.22)
Colon cancer [4]	29	1.09 (1.05–1.13)	1.24 (1.20–1.28)
Ovarian cancer [77]	34	1.06 (1.00–1.12)	NA
Stomach cancer [4]	8	1.04 (0.90–1.20)	0.97 (0.88–1.06)
Rectal cancer [4]	29	1.02 (1.00–1.05)	1.09 (1.06–1.12)
Later stage prostate cancer [73]	23	NA	1.08 (1.04–1.12)

NA = not available.

**Table 3 cancers-15-00485-t003:** Relationship between obesity and overall survival and cancer-specific survival in some major solid-organ cancers.

Type of Cancer [27]	Number of Cohorts	Overall Survival (HR, 95%CI)	Number of Cohorts	Cancer-Specific Survival (HR, 95%CI)
Breast	59	1.26 (1.20–1.33)	36	1.23 (1.15–1.32)
Colorectal cancer	30	1.22 (1.14–1.31)	13	1.24 (1.16–1.33)
Pancreas	6	1.36 (0.95–1.93)	3	1.28 (1.05–1.57)
Endometrial cancer	12	1.20 (1.04–1.38)	6	1.02 (0.75–1.39)
Prostate cancer	12	1.07 (0.91–1.25)	15	1.26 (1.08–1.47)
Gastroesophageal cancer	7	1.08 (0.77–1.52)	2	0.83 (0.58–1.16)
Bladder cancer	3	1.08 (0.98–1.20)	3	1.36 (0.96–1.93)
Hepatobiliary cancer	5	1.06 (0.89–1.25)	1	0.79 (0.50–1.24)
Ovarian cancer	4	1.03 (0.75–1.41)	4	1.06 (0.82–1.37)

**Table 5 cancers-15-00485-t005:** Current studies in cancer and obesity.

Registration Number	Phase	Title of the Study
NCT02750826	Phase III	Breast Cancer Weight Loss Study (BWEL Study)
NCT00719303	Phase III	Diet and Physical Activity Change or Usual Care in Improving Progression-Free Survival in Patients With Previously Treated Stage II, III, or IV Ovarian, Fallopian Tube, or Primary Peritoneal Cancer
NCT04447313	Phase III	Telephone Delivered Weight Loss, Nutrition, Exercise WeLNES Study
NCT05316467	Phase II/III	Weight Management Plus Megestrol Acetate in Early-stage Endometrioid Carcinoma
NCT00819208	Phase III	A Phase III Study of the Impact of a Physical Activity Program on Disease-Free Survival in Patients with High Risk Stage II or Stage III Colon Cancer: A Randomized Controlled Trial (CHALLENGE)
NCT04861220	Phase II/III	Surgical Pre-habilitation in Breast Cancer
NCT02730338	Phase III	INTense ExeRcise for surviVAL Among Men With Metastatic Prostate Cancer (INTERVAL-GAP4) (INTERVAL)
NCT04298086	Phase II	A Study of the Body’s Response to Exercise and a Plant-Based Diet in Overweight Postmenopausal Women With Breast Cancer
NCT04499950	Phase II	Adaptive Nutrition and Exercise Weight Loss (A-NEW) Study
NCT04931017	Phase II	Metformin for Chemoprevention of Lung Cancer in Overweight or Obese Individuals at High Risk for Lung Cancer
NCT05082519	Phase II	Caloric Restriction and Activity to Reduce Chemoresistance in B-ALL
NCT04717050	Phase II	Reducing Metabolic Dysregulation in Obese Latina Breast Cancer Survivors Using Physical Activity

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
