# Peer review of "Obesity and Cancer: A Current Overview of Epidemiology, Pathogenesis, Outcomes, and Management"

_cancers, 2023, doi:10.3390/cancers15020485_

Round 1
Reviewer 1 Report (Previous Reviewer 4)
Dear Editor,
Thanks for the re-invitation. I initially rejected this manuscript with my comments. I believe the authors have tried to justify the points I raised, but I am still not fully satisfied.
Reviewer 2 Report (Previous Reviewer 2)
The revised version of the review has been greatly improved and has addressed all my concerns. The language is clear and concise. I recommend that the review is accepted for publication without further revision.
I apologized for my mistake in my previous comments submission. The comments were factually for another manuscript.
Reviewer 3 Report (Previous Reviewer 1)
As already indicated in my former review, the manuscript is very well written, informative and extremely useful.
This manuscript is a resubmission of an earlier submission. The following is a list of the peer review reports and author responses from that submission.
Round 1
Reviewer 1 Report
Beautiful review - Well written, easy to read and quite instructive. The plan is well designed. The figures quite helpful.
Nothing specific to add
Reviewer 2 Report
I am aware of that most limitations of this study was due to the original data derived from the the Shanghai Link Healthcare Database, a real world clinical database covering more than 99% of Shanghai residents.
Generally, this is a well-written review with a deliberately designed structure. Particularly, the authors hightlighted the role of obesity in outcomes of cancer survivors, which indicte the importance of weight-reducing straeties and interventions in health care for obesity-associated cancers. The review can be improved in several aspects: 1) As this article is not a systematic review, I do not think it is necessary to present the strategy for literature search. Otherwise, the authors should provide more details on it. 2) The authors focused on components of "metabolic syndrome" in part 4, which was discussed as potential mechanisms underlying the roles of obesity in cancer. It would be better understood to combine the part 4 with the part 6. 3)It is perfect to summarize the related results derived from cohort studies in table 2 and table 3. However, it is unclear how the authors obtained the summarized RR and 95%CI, extracting from available meta-analysis or calculating by themselves?
Hope my comments help to improve the manuscript. Best regards,
Reviewer 3 Report
As it is a review article, the "Literature search" section is not satisfactory. There are several gaps that compromise the quality of the study. A well-designed review article, to be reliable, needs a good description of the methods. The authors used only PubMed, not describing the search equations or the search period. Eligibility criteria were not described.
Also, the writing has unscientific expressions, many references to support a statement, or sentences without references.
Reviewer 4 Report
This work is not a scoping or systematic review which is needed to make a contribution in scientific knowledge and to fill the research gaps. I have listed by concerns which are the basics of a review article to be published in scientific journals. In general, authors are highly encouraged to follow PISMA checklist of any other standard tool to conduct and report this review.
1. Identify the report as a systematic review in the title. So basically this is a very narrative review not a scoping or systematic review.
2. Please check if simple summary is needed, I think abstract is enough.
3. Rationale for the review in the context of existing knowledge is missing.
4. Only one search engine, PubMed is not enough to justify the exhaustive or systematic data search.
5. Data search prior to September 2022…..it means the studies 40 years ago might be part of this review which doesn’t justify the findings.
6.
7. Inclusion and exclusion criteria for the review and how studies were grouped for the syntheses is not available.
8. Provide the methods used to collect data from reports, including how many reviewers collected data from each report, whether they worked independently, any processes for obtaining or confirming data from study investigators, and if applicable, details of automation tools used in the process.
9. List of outcomes for which data were sought not available.